# Physiological and Molecular Responses of *Camellia oleifera* Seedlings to Varied Nitrogen Sources

**Rui Wang** [1,2,†], **Zhilong He** [1,2,†] , **Ying Zhang** [1,2], **Zhen Zhang** [1,2], **Xiangnan Wang** [1,2] and **Yongzhong Chen** [1,2,*]

1 Research Institute of Oil Tea Camellia, Hunan Academy of Forestry, Shaoshan South Road, No. 658, Changsha 410004, China; wangrui@hnlky.cn (R.W.); hezhilong2000@hnlky.cn (Z.H.); zhangying@hnlky.cn (Y.Z.); zhangzhen@hnlky.cn (Z.Z.); wangxiangnan@hnlky.cn (X.W.)
2 National Engineering Research Center for Oil Tea Camellia, Changsha 410004, China
* Correspondence: chenyongzhong@hnlky.cn
† These authors contributed equally to this work.

**Abstract:** *Camellia oleifera* Abel. is a unique woody edible oil tree species in China mainly distributed in red soil areas. Nitrogen is one of the important growth-limiting factors for *C. oleifera*, and the nitrogen form has a vital impact on the growth and development of the plants. Ammonium and nitrate are the two nitrogen forms that are maximally absorbed and utilized by the plants. Here, we used one-year-old seedlings of the main varieties of *C. oleifera* ('Xianglin No. 1', 'Xianglin No. 27', and 'Xianglin No. 210') to set up six experimental groups treated with different nitrogen forms ($[NO_3^- : NH_4^+]$ 0:0, 10:0, 7:3, 5:5, 3:7, and 0:10) and investigated the effects of nitrogen on the vegetative growth and photosynthetic characteristics of the seedlings. This study showed that mixed nitrogen sources could significantly increase the seedling height, ground diameter, biomass accumulation, and photosynthesis efficiency. Transcriptome sequencing analysis led to the identification of 3561 differentially expressed genes in the leaves. Preliminary screening identified several key enzyme genes from nitrogen nutrient metabolic pathways that were differentially expressed among seedlings grown with different nitrogen forms, and their expression pattern changes were further characterized. All the results demonstrate that the same proportion of ammonium nitrate ratio promoted the expression of genes encoding glutamine synthetase and glutamate synthase, thereby improving nitrogen assimilation and utilization efficiency. This study could provide a theoretical basis for scientific and rational fertilization and the improvement of nitrogen utilization efficiency in *C. oleifera* seedlings.

**Keywords:** oil tea camellia; nitrogen nutrient; seedling growth; photosynthetic characteristics; transcriptome analysis

## 1. Introduction

*Camellia oleifera* is a unique woody oil tree species in China and one of the four major woody edible oil tree species in the world [1]. Its main product, camellia oil, is a healthy edible oil, which is listed by the United Nations Food and Agriculture Organization as a healthy high-grade edible oil for key promotion [2]. In addition, oil tea and its by-products have a wide range of uses in nutrition and medicine, cosmetics, the agricultural industry, and other fields, and therefore, have high economic benefits.

Nitrogen is one of the essential nutrients required for plant growth and development, and it is an important component of nucleic acids, proteins, enzymes, and other metabolic products in plants. It is also an important factor that affects plant growth, development, and yield, and plays an irreplaceable role in plant life activities [3]. When lacking nitrogen, the synthesis of organic matter is hindered, resulting in dwarf plants, yellowing leaves, and reduced yield. On the other hand, when there is an excess of nitrogen, it leads to darker green leaves, excessive growth of branches and leaves, delayed maturity, decreased resistance to adverse environmental conditions, susceptibility to pests and diseases, underdeveloped mechanical tissue of stems, and lodging. When single ammonium nitrogen is applied,

$NH_4^+$ tends to destroy the membrane structure and induce oxidative phosphorylation and photosynthetic phosphorylation with the uncoupling of electron transfer [4]. Therefore, the proper use of nitrogen fertilizer plays a crucial role in plant growth. The main forms of nitrogen absorbed and utilized by plants are ammonium, nitrate, and organic nitrogen, with the largest absorption amount being ammonium and nitrate. Different forms of nitrogen have significant differences in their impact on plant growth and development, and also produce different physiological effects on plants [5].

Nitrogen is the most important nutrient that affects plant growth and development, and excessive or insufficient nitrogen fertilization could adversely affect the growth and yield of *C. oleifera* [6]. To promote the high yield of *C. oleifera*, extensive research has been conducted on the effects of nitrogen fertilizer application in *C. oleifera* forests. Currently, most research on *C. oleifera* fertilization techniques focuses on selecting different types of fertilizers such as nitrogen, phosphorus, and potassium, balanced fertilization, and formula ratios [7]. The research on nitrogen fertilizer application in *C. oleifera* has also focused on the impact of formula fertilizers on seedling growth. For example, a study found that the fertilization ratios of N/P/K = 12:4:4 and N/P/K = 10:6.7:3.3, respectively, had the best effects on young and mature *C. oleifera* forests [8]. Another study found that when the ratio of nitrogen, phosphorus, and potassium fertilizer applied was 1:2:2, the yield of *C. oleifera* was significantly increased. It was also found that applying nitrogen fertilizer could promote the nutritional growth of *C. oleifera*, while applying phosphorus fertilizer was beneficial for the differentiation of *C. oleifera* flower buds [9]. Through the above research, both the technology and effects of nitrogen fertilizer application in *C. oleifera* forests have been significantly improved.

Nitrate and nitrogen are the main nitrogen forms that are absorbed by plants from the soil, and they differ greatly in their physiological effects on plants [10]. In agriculture, there are many studies on the effects of nitrogen forms and their ratios on crops [11,12]. Terce-Laforgue et al. evidenced a higher relative amount of Rubisco of tobacco plant in ammonium than in nitrate nutrition [13]. This finding from gas exchange measurements showed that, compared to nitrate-supplied plants, ammonium-supplied plants had a high $CO_2$ assimilation rate under high $CO_2$ and/or high light supply. However, there is still a lack of reports on further exploration of the nitrogen level and utilization of nitrogen forms in *C. oleifera*, which, to some extent, still limits the scientificity of nitrogen fertilizer application in *C. oleifera*. Yuan et al. [14] found that applying nitrogen fertilizer at different levels can promote the characteristics of *C. oleifera* photosynthesis and improve soil fertility, with the level of effects being medium nitrogen > high nitrogen > low nitrogen. Chen et al. [15] found that different forms of nitrogen had a significant impact on the growth of *C. oleifera*. Among them, the growth of *C. oleifera* treated with ammonium was the largest, and the root-to-crown ratio and underground biomass were the highest. When treated with ammonium-nitrate mixture, the total fresh weight was the highest. The result of another study showed that uptake kinetic parameters *Vmax* $NH_4^+$ > *Vmax* $NO_3^-$ and *Km* $NO_3^-$ > *Km* $NH_4^+$, indicating that the uptake potential of ammonium–nitrogen by *C. oleifera* seedlings is greater than that of nitrate–nitrogen [16]. Therefore, in-depth research on the relationship between the absorption and utilization of different forms of nitrogen by *C. oleifera* and its growth and yield are of great theoretical and practical significance for targeted fertilization management in the field.

In previous studies, it has been demonstrated that *C. oleifera* exhibits varying growth performances in response to different nitrogen form ratios. Building upon this knowledge, our study aims to investigate the response mechanisms of *C. oleifera* seedlings to different nitrogen form ratios and determine the optimal nitrogen levels, forms, and ratios for promoting the growth of *C. oleifera* seedlings. We hypothesized that specific nitrogen form ratios will have significant effects on the nutritional growth of seedlings, root development, leaf photosynthetic characteristics, and the expression of related genes in leaves. By elucidating these effects, we anticipate providing valuable insights and technical support for the efficient utilization of nitrogen fertilizers in *C. oleifera* seedlings.

## 2. Materials and Methods

### 2.1. Overview of the Test Site

The test site was located in the National Oil-Tea Camellia Germplasm Resources Collection and Preservation Bank of the Hunan Academy of Forestry Sciences (National Oil-Tea Camellia Engineering & Technology Research Center Test Station). It is situated at 113° 01′ east longitude, 28° 06′ north latitude, and 80–100 m above sea level. This area has a subtropical monsoon climate, with a lot of rain in the late spring and early summer and drought in the summer and autumn. Generally, the annual average temperature is 16.8–17.3 °C, the extreme maximum temperature is up to 40.6 °C, and the extreme minimum temperature reaches −12 °C. The annual average rainfall is 1422 mm, the frost-free period is 275 days, and the annual average relative humidity is 80%. The soil is Quaternary red earth, with a pH value between 4.5 and 5.5, an organic matter content of 41.01 $g \cdot kg^{-1}$, a total nitrogen content of 2.68 $g \cdot kg^{-1}$, a total phosphorus content of 0.61 $g \cdot kg^{-1}$, and a total potassium content of 4.53 $g \cdot kg^{-1}$.

### 2.2. Experimental Materials

One-year-old seedlings of *C. oleifera* 'Xianglin No. 1' (XL1), 'Xianglin No. 27' (XL27), and 'Xianglin No. 210' (XL210) were used in this study. The seeds were provided by the Hunan Academy of Forestry Sciences (National Oil-Tea Camellia Engineering Technology Research Center). The three varieties of *C. oleifera* used in this study are all excellent varieties widely used in the planting industry of *C. oleifera*. The seedlings showed good growth and development in the early stage of growth, and the growth conditions were basically the same. In this experiment, we randomly selected three varieties of seedlings as experimental materials for research.

### 2.3. Experimental Design

Fertilizers were applied by liquid irrigation. To ensure that the seedlings grew under normal nutritional conditions, a modified Hoagland nutrient solution (nitrogen-free) was used for the fertilization, the formulation of the nutrient solution was: $K_2SO_4$ 261.39 $mg \cdot L^{-1}$, $KH_2PO_4$ 136.09 $mg \cdot L^{-1}$, $CaCl_2$ 221.98 $mg \cdot L^{-1}$, $MgSO_4 \cdot 7H_2O$ 246.47 $mg \cdot L^{-1}$, $MnSO_4 \cdot H_2O$ 1.54 $mg \cdot L^{-1}$, $H_3BO_3$ 2.86 $mg \cdot L^{-1}$, $ZnSO_4 \cdot 7H_2O$ 0.22 $mg \cdot L^{-1}$, $CuSO_4 \cdot 5H_2O$ 0.08 $mg \cdot L^{-1}$, $Na_2MoO_4 \cdot 2H_2O$ 0.02 $mg \cdot L^{-1}$, and $FeSO_4 \cdot 7 H_2O$ 20 $mg \cdot L^{-1}$. In addition, 7 $\mu mol \cdot L^{-1}$ nitrification inhibitor dicyandiamide ($C_2H_4N_4$) was added to all nutrient solutions to inhibit nitrification. The experiment was conducted using a completely randomized design. According to a previous study [17], six treatments (Table 1) were carried out, and each treatment was set up in triplicate, with 200 seedlings of each planted in a nutrient bowl with a height and diameter of 6.5 cm for each repeat. The first fertilization started in mid-June, followed by one fertilization every other week for a total of ten fertilizations. Each seedling was watered thoroughly with 300 mL of the nutrient solution. The solution collected in the trays was used for re-watering to ensure no loss of nitrogen. The experiments were conducted in a greenhouse at 20.0–25.0 °C with 6000–8000 lux of light and 80–85% relative humidity. During the experiment, in addition to the fertilization, normal watering, weeding, and other maintenance required, we kept the relative water content of the soil matrix between 75 and 80%.

**Table 1.** Nitrogen levels and ratios under various treatments.

| Treatment | Different Nitrogen Ratios of $m(NO_3^--N)/m(NH_4^+-N)$ | Total Nitrogen Content/mmol·L$^{-1}$ | Nitrogen Source/mmol·L$^{-1}$ | |
| --- | --- | --- | --- | --- |
| | | | NaNO$_3$ | (NH$_4$)$_2$SO$_4$ |
| A0(CK) | 0:0 | 0 | 0 | 0 |
| A1 | 10:0 | | 8 | 0 |
| A2 | 7:3 | | 5.6 | 1.2 |
| A3 | 5:5 | 8 | 4 | 2 |
| A4 | 3:7 | | 2.4 | 2.8 |
| A5 | 0:10 | | 0 | 4 |

### 2.4. Measurement of Seedling Growth Indicators

The investigation was conducted on June 15 (before fertilization) and November 15 (the end of seedling growing) in 2019. Thirty seedlings were randomly selected for each replicate of each treatment, and the seedling height was measured with a tape measure, and the ground diameter was measured with a vernier caliper. The vegetative growth was assessed using the formula: growth = W2 − W1. In the formula, W1 is the data collected prior to the start of the experiment, and W2 is the data collected in November. Sampling was carried out to determine the biomass. Fifteen seedlings were randomly selected for each replicate of each treatment, and the fresh weights of the above-ground parts (stems and leaves) and below-ground parts (roots) of the seedlings were weighed. Afterward, seedlings were placed in an oven at 105 °C for 30 min and then baked at 80 °C until they were completely dry. Finally, the dry weight was weighed.

### 2.5. Measurement of Root Morphological Indicators

After the seedlings stopped growing, ten differently-treated seedlings with the same growth were selected on November 15, and the process was repeated three times. The root system of the seedlings was scanned by a WanShen LA-S plant root system analyzer (Hangzhou Wanshen Detection Technology Co., Ltd., Hangzhou, China), and various morphological indicators of the root system were quantitatively analyzed. Root morphological parameters mainly included total root length, surface area, volume, and root projection area.

### 2.6. Measurement of Photosynthesis Indicators

Photosynthesis parameters were measured using an LI-6400 portable photosynthesis analyzer on a sunny morning in September 2019. Seedlings of almost the same growth were selected for the measurement, and five seedlings were selected for each treatment. The functional leaves located at the third to fifth leaf were selected for the measurement in triplicate. The photosynthetic photon flux density was set at 1000 μmol·m$^{-2}$·s$^{-1}$, the flow rate was set to 500 μmol·s$^{-1}$, the CO$_2$ concentration was set to 400 μmol·mol$^{-1}$, the leaf chamber temperature was 25 °C, and the measurement time for each gradient control was 120 s.

### 2.7. High-Throughput Sequencing and Differential Gene Expression Analysis

Total RNA samples from three biological replicates treated with the different nitrogen forms were prepared for transcriptome sequencing. cDNA library construction and sequencing were carried out by the Sangon Biotech (Shanghai) Co., Ltd. The clean reads of each sample were mapped to the assembled reference sequence, and the read count number of each sample compared to each gene was then obtained and converted to the expected number of fragments per kilobase of transcript sequence per millions of base pairs sequenced (FPKM). The gene expression level of each sample was estimated by RSEM and then analyzed. The read count data were used as the input data for the differential gene expression analysis, and the differential expression analysis among the samples/groups was performed using the DESeq2 package with a screening threshold of padj < 0.05 and

|log$_2$FoldChange| > 1. Finally, differentially expressed genes (DEGs) were identified by a two-by-two comparison. To obtain comprehensive gene function information, the DEGs were subjected to gene function annotation and clustering analysis based on nine databases, including CDD, KOG, Nr, Nt, Pfam, Swiss-prot, TrEMBL, KEGG, and GO.

### 2.8. Fluorescent Quantitative PCR Validation

Total RNA was extracted from *C. oleifera* leaf samples using the Easy Fast Plant Tissue RNA Rapid Extraction Kit (Tiangen Biochemical Technology Co., Ltd., Beijing, China). Reverse transcription and fluorescent quantitative PCR were performed using the PrimeScript™ RT reagent Kit with gDNA Eraser (Perfect Real Time) and TB Green® Premix Ex Taq™ (Tli RNaseH Plus) kit from Takara Biomedical Technology (Beijing) Co., Ltd. (Beijing, China), respectively. The initiation factor gene *ETIF3H* was included as an internal reference gene. The expression levels of key genes involved in nitrogen metabolism and transport in five *C. oleifera* seedlings were analyzed by the $2^{-\Delta\Delta CT}$ method [18]. The sequences of primers used in this study are shown in Table 2.

**Table 2.** Primers used for fluorescent quantitative PCR.

| Gene | Forward Sequence | Reverse Sequence |
|:---:|:---:|:---:|
| *NR* | CTCCATCCATCGTCACCTCTAC | TAACTCAGTAATCACCACACCTTG |
| *NiR* | GCCTCTTGATGCCTGGGTTC | TTCTTGTCTTCTGCCTGTGTCC |
| *GOGAT* | CTCCTCCAAACCCTTCCTACC | CAATGTCATCGCCTCTCACTG |
| *GS* | TCCTAATTCCTCCGCTGATTCTTC | AGTCGCCACCTTCAACAACC |
| *ETIF3H* | GACACCTTGGGAGGACTTTG | GTGGATAATAATACTGTTGGATGG |

### 2.9. Data Analysis

The correlation and principal component analysis of the growth and photosynthesis of *C. oleifera* seedlings under different nitrogen forms were carried out using R-studio and corresponding R package. The data were processed and statistically analyzed using Excel 2007 and SPSS25.0 software. One-way ANOVA and LSD were employed to compare the differences among the different treatments, and a *p*-value < 0.05 was considered statistically significant. Data plotting was carried out using GraphPad Prism 8.0.

## 3. Results

### 3.1. Effects of Different Nitrogen Forms on the Vegetative Growth of C. oleifera Seedlings

In order to study the effects of different nitrogen forms and combination treatments on nutrient growth, we measured the seedling height, ground diameter, above-ground biomass, below-ground biomass, and total biomass of *C. oleifera* seedlings in each treatment group. Among the five nitrogen form treatment groups (A1–A5), A3 displayed the largest increment of seedling height at 10.53 cm, which was 102.5% higher than that of the control group (Figure 1A). The mixed nitrogen treatment groups (A2–A4) did not significantly differ in height, but A3 had a significantly larger increment in height than each of the single nitrogen treatment groups (A1 and A5). Notably, the increment of seedling height was significantly higher in all the treatment groups (A1–A5) than in the control group (A0). Likewise, A3 had the largest increment in ground diameter of the seedlings among the five groups. While there was no significant difference in the increments between A3 and A2, A3 displayed a significantly larger increment than the other three nitrogen treatment groups (A1, A4, and A5). The increments were significantly higher in all five treatment groups (A1–A5) than in the control group. Moreover, A3 had significantly larger above-ground and total biomasses than the other four treatment groups (Figure 1B). In comparison, there was no significant difference in the below-ground biomass between A3 and A4, while the biomass was significantly larger in A3 than in A1, A2, and A5. Strikingly, A3 displayed the highest value of total biomass (2.54 g) at a $NO_3^-/NH_4^+$ of 5:5, which was 67.11% higher than the control group.

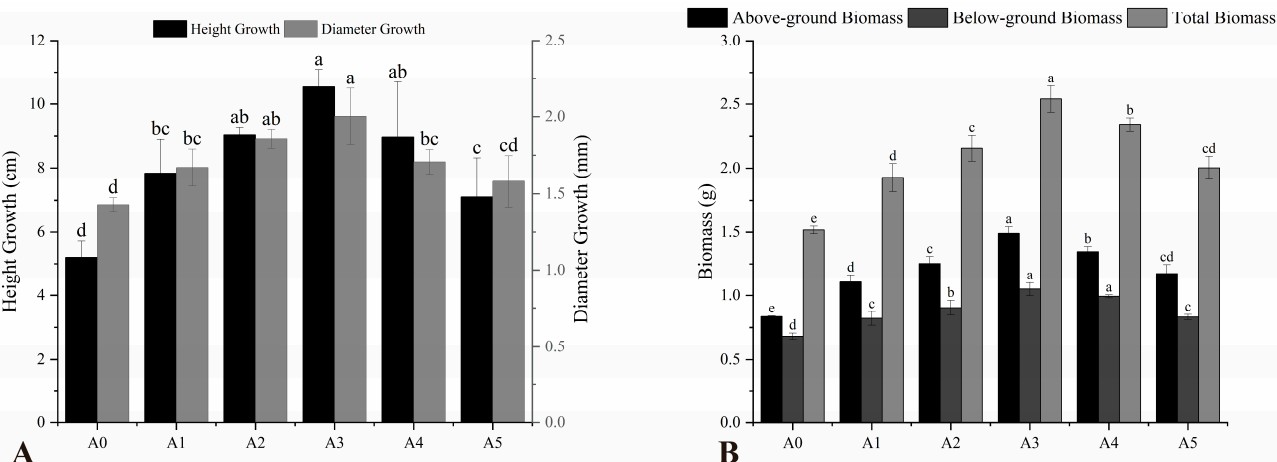

**Figure 1.** Effects of different nitrogen forms on the vegetative growth of *C. oleifera* seedlings. (**A**) Seedling height and diameter growth of different treatment; (**B**) seedling above-, below-ground, and total biomass of different treatment. Each data point represents the average of three biological replicates. Error bars represent mean ± SE. Different lowercase letters indicate significant differences at $p < 0.05$ by LSD method.

### 3.2. Effects of Different Nitrogen Forms on Root Growth of C. oleifera Seedlings

In order to study the effects of different nitrogen forms and combination treatments on root growth, the root length, root volume, root projection area, and root surface area of *C. oleifera* seedlings were measured. The nitrogen forms affected the root length of *C. oleifera* seedlings, and with the increase in the ammonium nitrogen ratio, the root length showed an inverted "V" trend (Figure 2A). Among the five nitrogen form treatment groups, A3 had the highest root length ($1.061 \times 10^3$ cm), which was 69.22% higher than that of the control group (A0). In contrast, A5 displayed the shortest root length ($0.429 \times 10^3$ cm), which was 12.41% lower than that of the control group. There was a significant difference in root length between A5 and the control group. The root volume of the seedlings was increased first and then decreased with the increase in the ammonium nitrogen ratio. The root volume was higher in the two mixed nitrogen treatment groups (A3–A4) than in the single nitrogen treatment (A1 and A5) and control groups (A0). Notably, A3 had the largest root volume (16.01 cm$^3$) among all the treatment groups, which was 118.42% larger than that of the control. A4 exhibited the second largest root volume (15.24 cm$^3$), which was 107.91% larger than that of the control. There was no significant difference in the root volume between A4 and A3. Like the root volume, the root projected area and surface area of the seedlings were increased first and then decreased with the increase in the ratio of ammonium nitrogen (Figure 2B). The root projection area and surface area were larger in the mixed nitrogen treatment groups than in the single nitrogen treatment and control groups. Among all the treatment groups, A3 had the largest root projection area (90.79 cm$^2$), and A5 had the smallest root projection area (45.79 cm$^2$), which was smaller than that of the control group. Strikingly, A3 displayed the largest root surface area (285.91 cm$^2$), which was significantly larger than that of the other treatment (A1, A2, A4, and A5) and control groups (A0). On the contrary, A5 had the smallest root surface area (141.95 cm$^2$).

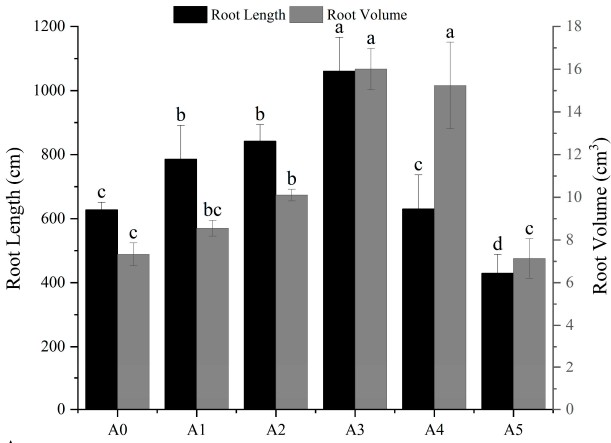
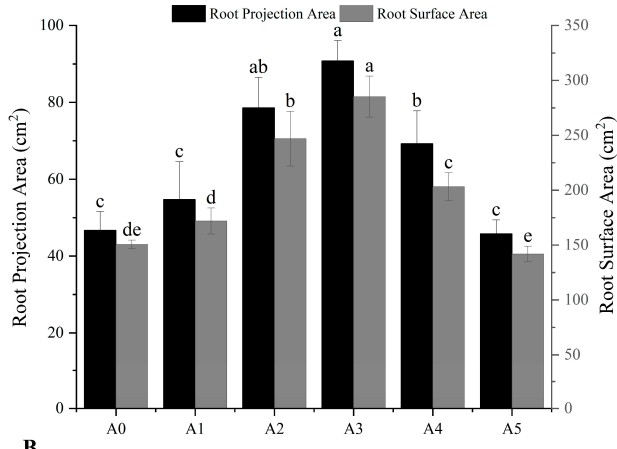

**Figure 2.** Effects of the different nitrogen forms on the root growth of *C. oleifera* seedlings. (**A**) Root length and volume of different treatments; (**B**) root projection and surface area of different treatments. Each data point represents the average of three biological replicates. Error bars represent mean ± SE. Different lowercase letters indicate significant differences at *p* < 0.05 by LSD method.

### 3.3. Effects of the Different Nitrogen Forms on the Photosynthetic Characteristics of C. oleifera Seedlings

In order to study the effects of different nitrogen forms and combination treatments on photosynthesis, the net photosynthetic rate (Pn), stomatal conductance (gs), intercellular $CO_2$ concentration (Ci), and transpiration rates (E) of *C. oleifera* seedlings were measured. The Pn of *C. oleifera* leaves was increased first and then decreased with the increase in the ammonium nitrogen ratio (Figure 3A). Among the five nitrogen form treatment groups, A3 had the highest Pn (5.62 $\mu mol \cdot m^{-2} \cdot s^{-1}$), which was not significantly different from that of A4 but was significantly higher than that in the other treatment groups (A1, A2, and A5). In contrast, the control group (A0) had the lowest Pn (3.58 $\mu mol \cdot m^{-2} \cdot s^{-1}$). Compared to the control, the Pn values of the A1–A5 groups were increased by 30.25%, 31.62%, 56.98%, 50.06%, and 13.54%, respectively. The Pn was higher in the mixed nitrogen treatment groups (A2–A4) than in the single nitrogen treatment groups (A1 and A5), and the full nitrate treatment group (A1) had a higher Pn than the full ammonium treatment group (A5). The gs of the leaves showed a consistent change trend with the Pn (Figure 3B). A3 displayed the highest gs (0.36 $mol \cdot m^{-2} \cdot s^{-1}$) among all the treatment groups, and no significant differences in the gs were observed among the treatment groups (A1–A5). Each of the five treatment groups had a higher gs than the control group, and the gs at a $NO_3^- \text{-N} / NH_4^+ \text{-N}$ of 5:5 was significantly higher in A3 than in the control group. In this study, there was no significant difference in the Ci index of *C. oleifera* seedlings leaves between the treatment group and the control group, nor between each treatment group (Figure 3C). With the increase in the ammonium nitrogen ratio, the E of the leaves exhibited a consistent change trend with the Pn (Figure 3D). The E was higher in the mixed nitrogen treatment groups than in the single nitrogen treatment groups. Among all the treatment groups, A3 had the highest E (3.17 $mmol \cdot m^{-2} \cdot s^{-1}$), and A0 had the lowest E (1.97 $mmol \cdot m^{-2} \cdot s^{-1}$). Compared to the control, the E values of the A1–A5 groups were increased by 35.09%, 54.02%, 60.26%, 58.88%, and 26.78%, respectively. The instantaneous carboxylation efficiency (IcE) was calculated, and the results show that A3 and A4 had the highest IcE.

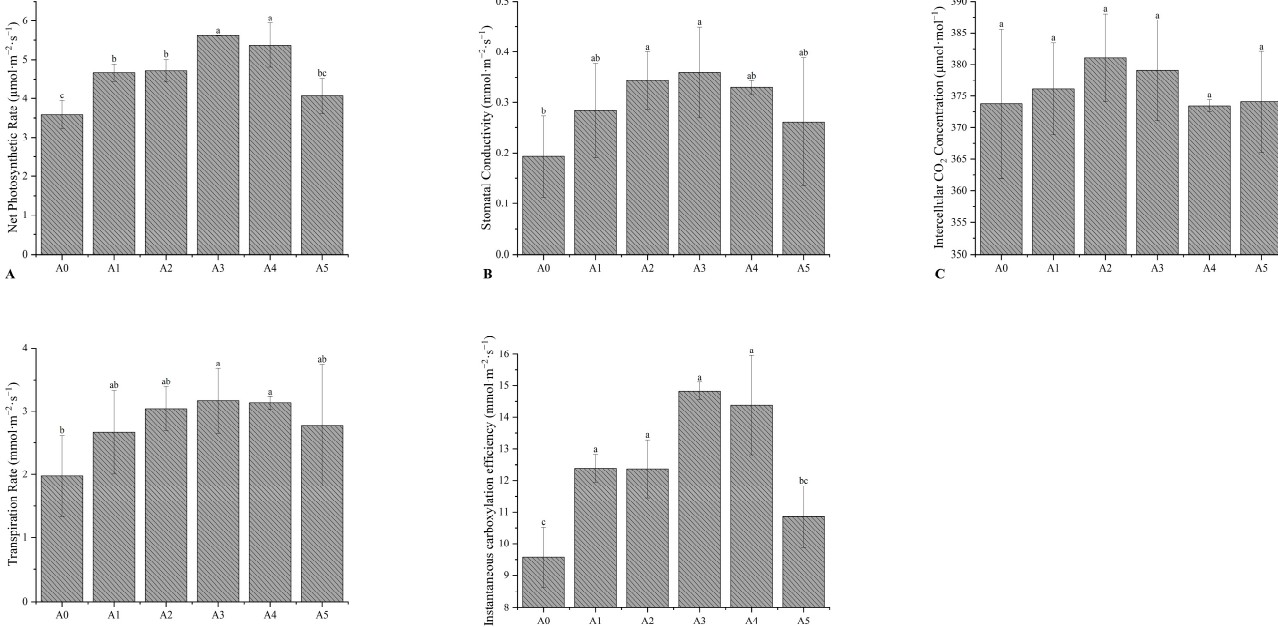

**Figure 3.** Effects of the nitrogen forms on the photosynthesis of *C. oleifera* seedlings. (**A**) Net photosynthetic rate; (**B**) stomatal conductivity; (**C**) intercellular $CO_2$ concentration; (**D**) transpiration rate; (**E**) instantaneous carboxylation efficiency. Each data point represents the average of three biological replicates. Error bars represent mean $\pm$ SE. Different lowercase letters indicate significant differences at $p < 0.05$ by LSD method.

### 3.4. Principal Component Analysis of the Effect of Nitrogen on C. oleifera Seedlings

The correlation analysis and principal component analysis of seedling growth-related indicators identified significant positive correlations among the physiological indicators reflecting the growth of the seedlings and roots and photosynthetic parameters, such as the Pn (Figure 4). The contribution rates of the top three principal components were 61.1, 18.1, and 9.9%, with a cumulative contribution rate of 89.1%. The first principal components mainly included physiological indicators reflecting the growth of the seedlings and roots, and the Pn, gs, and E. Meanwhile, the second principal components mainly contained the gs, E, and intercellular carbon dioxide concentration. The distribution analysis of the first and second principal components of the seedlings in each treatment group show that the first principal component distribution was relatively concentrated among the individuals in each treatment group, and this distribution pattern well reflects the overall growth of the seedlings in each treatment group. All these results indicate that the treatment in A3 was the most favorable for the overall growth of the seedlings, followed by the treatments in A2 and A4, and the treatment in A0 resulted in the worst outcome.

### 3.5. Differentially Expressed Genes under Different Nitrogen Forms

Among the five treatment groups, A1 had the lowest number of DEGs compared to the control group (Table 3). In contrast, A3 had the greatest number of DEGs (1883) in the leaves compared to the control group; among the 1883 genes, 1024 were up-regulated, and 859 were down-regulated. In A2 and A4, the number of up-regulated DEGs was much greater than that of the down-regulated DEGs in the leaves.

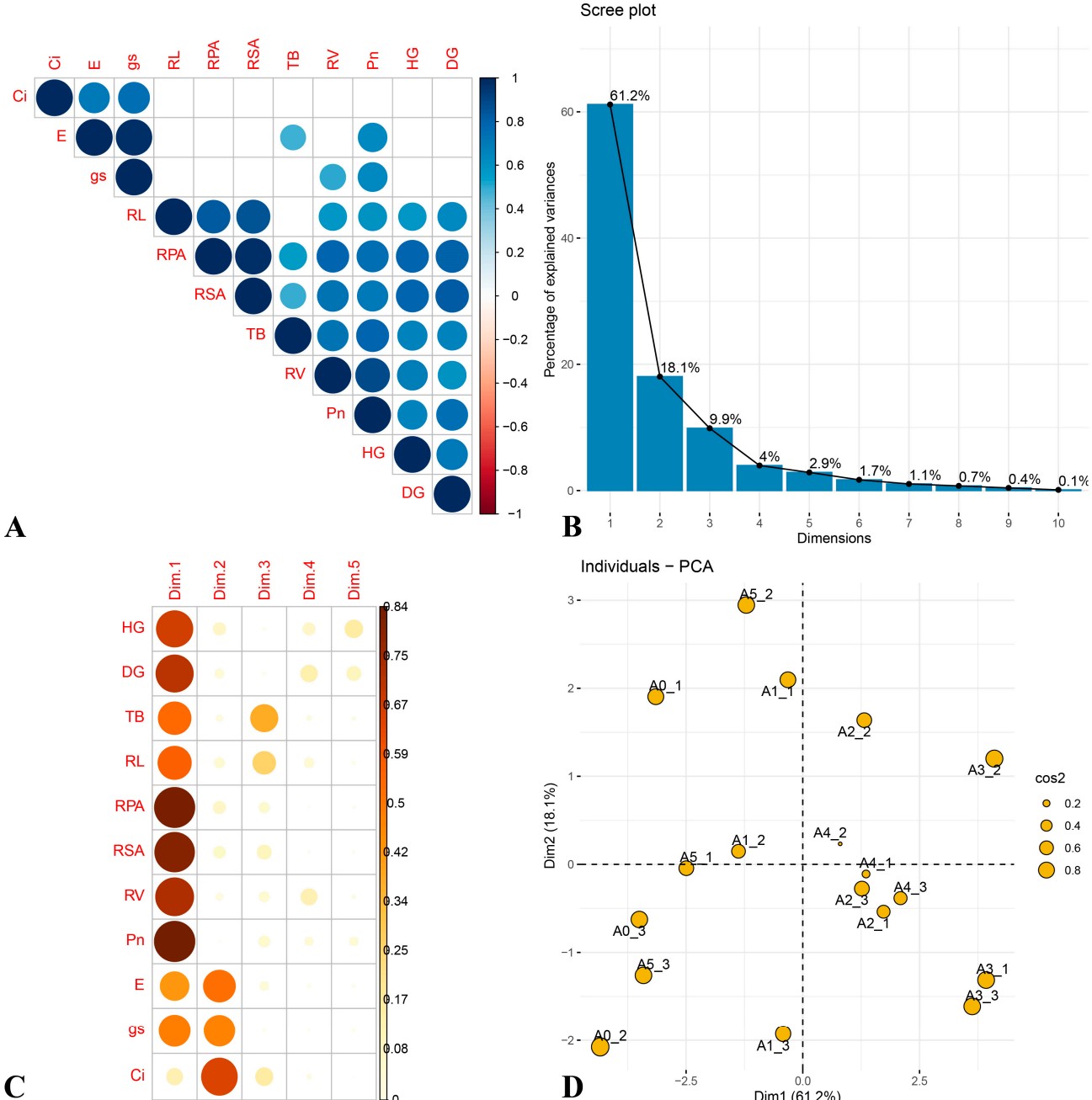

**Figure 4.** The correlation analysis and principal component analysis of the seedling growth-related indicators. (**A**) correlation analysis of the seedling growth-related indicators; (**B**) variance contribution rate of each principal component; (**C**) correlation between seedling growth-related indicators and principal components; (**D**) visualization of principal component analysis results. HG: height growth; DG: diameter growth; TB: total biomass; RL: root length; RV: root volume; RPA: root project area; RSA: root surface area.

**Table 3.** Statistical analysis of DEGs in the seedling leaves under different nitrogen forms.

| Regulation | A1_vs_A0 | A2_vs_A0 | A3_vs_A0 | A4_vs_A0 | A5_vs_A0 |
|---|---|---|---|---|---|
| Up | 45 | 304 | 1024 | 829 | 486 |
| Down | 72 | 148 | 859 | 225 | 444 |
| Total | 117 | 452 | 1883 | 1054 | 930 |

The number of shared and unique DEGs among the groups was analyzed using a Venn diagram. As illustrated in Figure 5, among all comparison groups, A3l vs. A0l had the greatest number of unique up-regulated DEGs (773), followed by A4l vs. A0l (627). In addition, the highest number of shared up-regulated DEGs (98) was found in A3l vs. A0l and A4l vs. A0l, while A3l vs. A0l and A5l vs. A0l had the highest number of shared down-regulated DEGs (97).

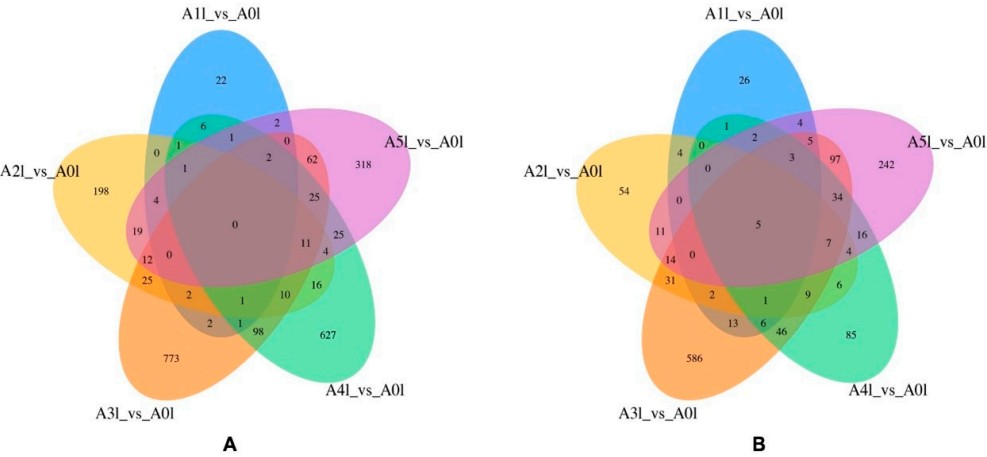

**Figure 5.** Venn diagram of up-regulated (**A**) and down-regulated (**B**) DEGs in the leaves under different nitrogen forms.

A GO enrichment analysis was performed on the DEGs of the major subclasses of leaves under the different nitrogen forms. The DEGs were mainly enriched in biological processes such as reproduction, reproductive process, developmental process, multicellular organismal process, response to stimulus, and rhythmic process. In terms of molecular function, the DEGs were mainly enriched in catalytic activity and binding. The main cellular components of the DEGs included symplast, cell junction, membrane, and membrane part.

A gene function analysis based on the KEGG system helps to investigate genes and their expression data as a whole network. A KEGG enrichment analysis revealed that 30 DEGs were annotated to signal transduction, followed by 27 for carbohydrate metabolism, 23 for energy metabolism, and 20 for lipid metabolism.

### 3.6. Effects of the Different Nitrogen Forms on the Nitrogen-Related Genes

Analysis of the transcript levels by transcriptome sequencing showed that the expression of the nitrate reductase was only slightly up-regulated in the leaves (Figure 6A). The real-time fluorescence quantitative PCR assay revealed that the expression of *Nitrate Reductase* (*NR)* gene in the leaves was significantly up-regulated in all the nitrogen-supplemented groups compared to the control group. Additionally, the expression of the *NR* in the leaves was significantly down-regulated (Figure 6B). The real-time fluorescence quantitative PCR assay showed that the expression level of *Nitrite reductase* (*NiR*) gene in the leaves was up-regulated in A1 and A2 compared to the control group. However, the expression of *NiR* was significantly down-regulated with the increase in the ammonium nitrogen ratio.

The expression of the *glutamate synthase* gene (*GS*) was up-regulated in the A4 group, while no significant differences in the expression levels of *GS* were observed among the other nitrogen treatment groups (A0, A1, A2, A3, and A5) (Figure 6C). The real-time fluorescence quantitative PCR assay had consistent results with the transcriptome sequencing. In this case, the expression level of *GS* was significantly up-regulated in A4. Particularly, the *GS* expression level in A4 was 2.03-fold and 2.48-fold higher than that in the control group, as revealed by the transcriptome sequencing and fluorescence quantitative PCR assay, respectively. The expression of the *glutamine synthetase* gene (*GOGAT*) in the leaves was up-regulated in the nitrogen-supplemented groups compared to the control group (Figure 6D). Real-time fluorescence quantitative PCR assay revealed that the expression of

*GOGAT* in the leaves was up-regulated in A1, A2, and A3, while it was down-regulated in A4 and A5. Notably, A3 displayed the highest expression level of *GOGAT* among all the treatment groups.

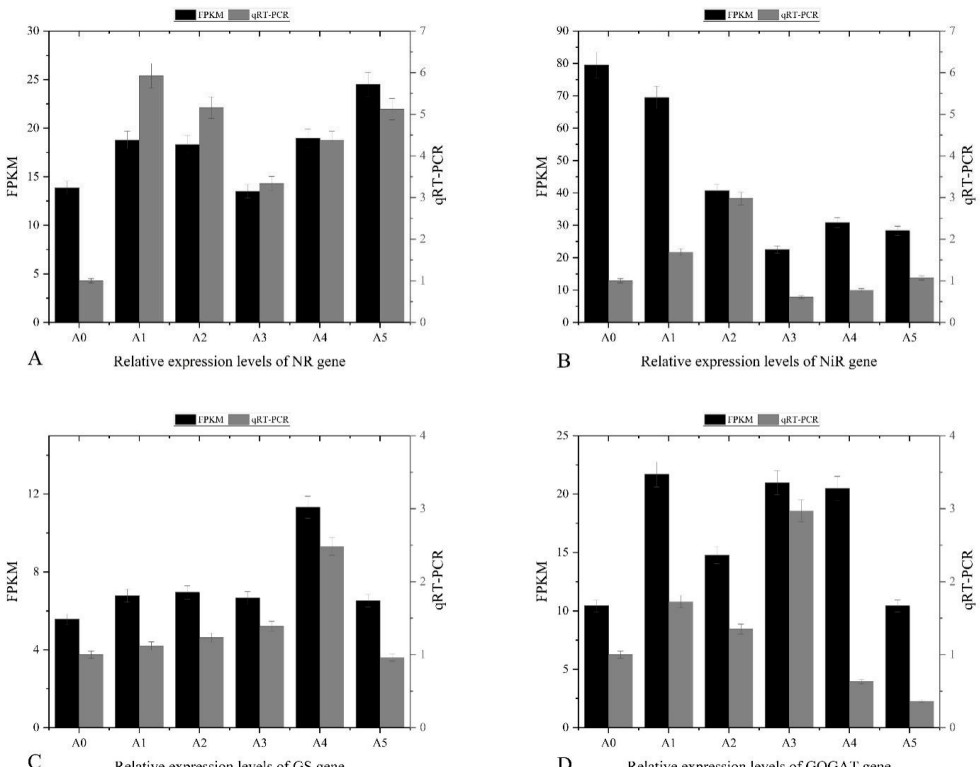

**Figure 6.** The expression levels of nitrogen metabolism-related genes in the leaves under different nitrogen forms. (**A**) the expression levels of *NR* gene; (**B**) the expression levels of *NiR* gene; (**C**) the expression levels of *GS* gene; (**D**) the expression levels of *GOGAT* gene.

## 4. Discussion

The growth status of seedlings can most intuitively reflect the nitrogen uptake of the seedlings, and distinct nitrogen forms significantly impact the growth and development of plants. Most plants grow more productively when nitrate nitrogen and ammonium nitrogen are applied simultaneously than when each single nitrogen form is supplied [11,19]. Moreover, the growth of plants exhibits a "combined effect" in which a certain proportion of nitrate nitrogen and ammonium nitrogen can increase the growth, yield, and new shoot length of plants. It has been shown that the addition of $NO_3^-$ to $NH_4^+$ alleviates the metabolic dysregulation caused by $NH_4^+$, while the addition of $NH_4^+$ to $NO_3^-$ reduces the large amount of reducing power and light quantum energy consumed by higher concentrations of $NO_3^-$ [20]. Consistently, the present study found that the mixed nitrogen source was conducive to enhancing the growth of seedlings, and the seedlings grew faster as the proportion of ammonium nitrogen in the mixed nitrogen source increased. Notably, the growth rate of seedlings decreased when the ratio of ammonium nitrogen exceeded 50% or 70%. In terms of the mechanism of promoting effects of the mixed nitrogen source on the seedling growth, the mixed nitrogen source may help to maintain the pH value of the cultivation medium, thus promoting the physiological metabolism, growth, and development of plants. Alternatively, $NO_3^-$ acts as a signaling molecule to induce the production of cytokinin and regulates the distribution of dry matter in the plant, thereby enhancing plant growth [21].

The root system is an important organ for seedlings to obtain water and nutrients. The length, volume, projection area, and surface area of the roots are indicators of root morphology, and nitrogen has a significant regulatory effect on root growth [22]. It has been

demonstrated that nitrogen-deficient nutrition leads to an increase in the biomass allocated to the root system, thus improving the nitrogen uptake capacity [23]. In contrast, sufficient nitrogen supply can cause an increase in the root diameter and surface area and a decrease in the specific root length. It has also been found that excessive nitrogen supply results in an inhibition of plant root growth. Studies have shown that there is a complementary effect between $NO_3^-$ and $NH_4^+$ [24]. $NH_4^+$ promotes the branching of lateral roots, while $NO_3^-$ induces lateral root elongation. The combined use of $NO_3^-$ and $NH_4^+$ can enhance the branching and elongation of lateral roots, thereby increasing the absorption capacity of the root system and promoting the above-ground growth of plants [25].

In this study, we showed that nitrogen in different forms and ratios significantly impacted root morphological indicators, including the root length, volume, projection area, and surface area. Moreover, both mixed nitrogen sources and the full nitrate treatments significantly increased the root length, surface area, and volume of the seedlings; all the values reached their maximum in the A3 treatment group ($[NO_3^-/NH_4^+]$ = 5:5). These findings are consistent with the results of studies on citrange seedlings. Given that the root length and root surface area are closely related to the nutrient absorption capacity of plants [26], the response of roots of the *C. oleifera* seedlings to nitrogen helps to improve the root system's access to water and nutrients in the soil.

Nitrogen forms affect almost all aspects of photosynthesis [27,28]. This study showed that mixed nitrogen sources could improve photosynthesis in leaves, and a single application of nitrate nitrogen was more conducive to increasing the Pn of leaves than a single application of ammonium nitrogen. Consistently, a study on the effect of the nitrogen form on the Pn in wheat found that plants grown with a mixed nitrogen source had the highest Pn, followed by those grown with single nitrate nitrogen and those with single ammonium nitrogen [29]. Furthermore, Song et al. [30] demonstrated that rice grown with mixed ammonium and nitrate nitrogen ($[NO_3^-: NH_4^+]$ = 3:2) displayed the highest chlorophyll content, photosynthetic rate, and light absorption capacity of the leaves, followed by those grown with full nitrate nitrogen, while all the indicators were lowest in the rice grown with full ammonium nitrogen. Studies have found that different nitrogen sources had different effects on the carbon and nitrogen metabolism pathways of *Camellia sinensis*. When $NO_3^-$ is utilized as a nitrogen source, there is an observed increase in the accumulation of tea polyphenols and catechins, which are representative of carbon pools. Concurrently, under the influence of the $NO_3^-$ treatment, the expression of genes related to catechin biosynthesis is up-regulated, suggesting a positive correlation between the $NO_3^-$ treatment and the enhancement of catechin biosynthesis. $NH_4^+$ and mixed nitrogen treatments were found to enhance the accumulation of amino acids, especially theanine, glutamic acid and arginine, which contribute to the freshness of *C. sinensis* [31]. The difference in the photosynthesis rate caused by different forms of nitrogen is related to the distinct metabolic processes of $NH_4^+$ and $NO_3^-$ in plants. In this case, the high concentration of $NH_4^+$ exerts a toxic effect on plants, reducing the photosynthesis rate. In the case of single nitrate nitrogen, the excess $NO_3^-$ is stored in the vacuole. As a result, the nitrogen does not affect plant metabolism and exerts no toxic effects on the plants.

A previous study showed that *C. oleifera* seedlings receiving a solution with both $NO_3^-$ and $NH_4^+$ at a ratio of 1:1 had the highest total N in leaves and total dry weight, elevated chlorophyll, soluble saccharide, and protein contents, as well as higher activities of peroxidase, superoxide dismutase, nitrate reductase, glutamine synthetase, and glutamate synthase [17,32]. In this study, we observed that the supply of single ammonium nitrogen led to a reduction in the photosynthetic capacity of the seedling leaves. This observation can explain, to a certain extent, why seedlings supplied with single nitrate nitrogen had a higher Pn in the leaves than those with single ammonium nitrogen. Another reason could be that the supply of single nitrate nitrogen produces a high level of NR activity in the leaves. Subsequently, high NR activity can rapidly reduce $NO_3^-$ and avoid the excessive accumulation of $NO_3^-$ while it promotes photorespiration, contributing to the synthesis of amino acids and proteins and an increased photosynthetic rate [33]. In this

study, both A3 and A4 had a higher Pn than the other nitrogen treatment groups. By combining these results with transcriptome sequencing data, we discovered a large number of differentially expressed genes under treatment A3. Through functional annotations, such as KEGG, many of these genes were found to participate in the photosynthesis pathway. However, the specific regulatory mechanisms of these genes still require further analysis. This comprehensive study provides a deeper understanding of the molecular responses under treatment A3 and lays the groundwork for future research.

A principal component analysis is a statistical method that combines multiple indicators and transforms them into a small number of comprehensive indicators for the purpose of simplification. Herein, we used a principal component analysis to investigate the effects of the nitrogen form and ratio on the growth indicators of *C. oleifera* seedlings. The analysis revealed that while all the nitrogen treatments promoted the growth of the seedlings, the mixed nitrogen sources (A3, A4, and A2) had a greater growth-promoting effect on the seedlings than the single nitrogen sources (A1 and A5) and the control (A0). The growth-promoting effect of nitrogen in different forms and ratios on the seedlings was ranked from strong to weak: A3 > A4 > A2 > A1 > A5 > A0.

Nitrogen is a component of many important substances in plants, such as proteins, enzymes, chlorophyll, nucleic acids, vitamins, and hormones [34,35]. Additionally, nitrogen has significant effects on chlorophyll synthesis, photosynthetic rate, dark reactions, and the main enzymes of photorespiration in crops while directly or indirectly affecting the growth and development of the plants. However, the availability of N is limited in the soil of subtropical ecosystems, and it is important to further elucidate the intrinsic mechanism of *C. oleifera* seedlings in absorbing and utilizing different ratios of nitrogen nutrients. The RNA-sequencing of *C. oleifera* roots treated with three synthetic forms of N, namely ammonium, nitrate, and ammonium nitrate ($NH_4NO_3$) revealed that the expression of several ammonium and nitrate transporters increased slightly in the $NO_3^-$-treated group, and significant changes were observed in the $NH_4^+$- and $NH_4NO_3$-treated groups [36]. The opposite expressions of some key N transport and assimilation genes under the different N treatments suggest that *C. oleifera* might have different N utilization pathways for the different N forms. These findings will help improve the N utilization efficiency of *C. oleifera* and guide its agricultural production in the future. In this study, we set up five treatment groups along with one control group and performed a differential expression analysis of the transcriptome in the seedling leaf and root tissues among these groups. A large number of DEGs were identified in this study. Notably, the seedlings grown with full nitrate nitrogen had the lowest number of DEGs in the leaves, indicating that the nitrogen had the lowest effect on the seedlings, presumably due to the very low uptake and utilization efficiency of nitrate nitrogen in *C. oleifera*. With the increase in the ammonium nitrogen ratio, the number of DEGs in the leaves reached the maximum at the $NO_3^-$/$NH_4^+$ ratio of 5:5, showing that the regulatory effect of different forms of nitrogen on the leaves was synergistically regulated by nitrate nitrogen and ammonium nitrogen. In this study, the gas exchange results show that A3 had the highest photosynthetic efficiency, and correspondingly, a large number of genes involved in the photosynthesis pathway obtained from the transcriptome sequencing data were differentially expressed under the A3 treatment. Generally, plants have specific uptake and utilization mechanisms for different nitrogen forms. After entering the roots, nitrate nitrogen is mainly transported from the xylem to the aboveground in the form of $NO_3^-$ and is assimilated by *NR* and *NiR* in the leaves [37]. Alternatively, the nitrogen is stored in the vacuole of root and leaf cells and then converted into proteins through a series of reduction processes. NR is the first and the rate-limiting enzyme in the reduction processes. Upon entering the roots, $NH_4^+$ is rapidly assimilated into amino acids and phthalides through pathways such as the glutamine synthetase–glutamate synthase (GS-GOGAT) pathway and then transported above-ground [38].

In this study, we used a KEGG enrichment analysis to preliminarily examine the DEGs involved in important metabolic pathways, such as carbon/nitrogen metabolism and phytohormone signaling pathways in leaves under different nitrogen forms. The

analysis showed that genes involved in carbon metabolism, photosynthesis, and chlorophyll synthesis in the leaves were significantly up-regulated in the A3 group, indicating that mixed nitrate nitrogen and ammonium nitrogen at the ratio of 5:5 may be the most favorable for photosynthesis in the seedlings. This is justified by the finding that A3 had the highest Pn among all the treatment groups. We also observed that genes involved in metabolic pathways, such as nitrogen metabolism and amino acid synthesis in the leaves were significantly up-regulated in A3, indicating that with the increase in the ammonium nitrogen ratio, related pathways, such as the nitrogen utilization pathway in leaves were activated by an appropriate proportion of nitrogen. Conversely, the excessive proportion of ammonium nitrogen inhibited the root uptake of nitrogen nutrients to a certain extent. This observation is consistent with the results that the full ammonium nitrogen treatment inhibited the root morphology of *C. oleifera* seedlings. This study identified several key enzyme genes of nitrogen nutrition metabolic pathways that were differentially expressed among the seedlings grown with different nitrogen forms and preliminarily analyzed the characteristics of their expression patterns. After synthesizing the results with the influence of various nitrogen forms on the growth, maturation, physiology, and biochemistry of the seedlings, it was ascertained that a 5:5 ratio of $NO_3^-$ to $NH_4^+$ could potentially be the most effective for *C. oleifera*. A suitable balance of nitrogen nutrients can stimulate the expression of genes encoding glutamine synthetase and glutamate synthase, thereby enhancing nitrogen assimilation and utilization efficiency. Collectively, these findings offer a theoretical foundation for the dissection of molecular mechanisms involved in the absorption and utilization of diverse nitrogen forms in *C. oleifera*.

**Author Contributions:** Software, Z.H.; validation, Y.Z.; investigation, R.W. and X.W.; data curation, Z.Z.; writing—original draft preparation, Z.H. and R.W.; writing—review and editing, Z.H. and R.W.; supervision, Y.C.; project administration, Y.C.; funding acquisition, Y.C. All authors have read and agreed to the published version of the manuscript.

**Funding:** This research was funded by the Seed Industry Innovation Project of Hunan Province (2021NK1007), the Major Special Project of the Changsha Science and Technology Bureau (KQ2102007), and the Natural Science Foundation of Hunan Province (Grant No. 2022JJ30325).

**Data Availability Statement:** The data presented in this study are available on request from the corresponding author.

**Acknowledgments:** All authors thank Sangon Biotech (Shanghai) Co., Ltd. for the efficiently high-throughput sequencing project.

**Conflicts of Interest:** The authors declare no conflict of interest.

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
