# Peer review of "Physiological and Molecular Responses of Camellia oleifera Seedlings to Varied Nitrogen Sources"

_horticulturae, doi:10.3390/horticulturae9111243_

Round 1

Reviewer 1 Report

Comments and Suggestions for Authors

The manuscript entitled “Physiological and molecular responses of Camellia oleifera seedlings under different nitrogen forms" is based on original research experiment and the presented results therein broaden the knowledge of plant sciences. To elucidate the response mechanism of C. oleifera seedlings to nitrogen, determine the nitrogen level, nitrogen form, and ratio suitable for the seedling growth, and provide technical support and theoretical basis for highly efficient utilization of nitrogen fertilizers for the seedlings were the main aims of the work. Authors conducted experiment in field conditions, during which physiological and morphological features were measured. High-throughput sequencing, differential gene expression analysis and fluorescent quantitative PCR validation were also performed. There is no doubt that this work is in the scope of Horticulturae journal. The publication presents some interesting studies. The work delivers some interesting results and can be important source of valuable information.

The introduction is properly composed. The materials and methods section contains the basic requested elements and provide information about the experimental preparations, analyses. The data analysis is generally properly provided. The results show valuable information. The obtained data are discussed sufficiently.

However, the authors made some shortcomings that must be corrected before the publication of the work:

1)      Authors did not make a research hypothesis.

2)      Keywords: they cannot be this same as in title.

3)      MM section, statistic analysis: Have all the criteria been met for AVOVA to be used? equality of variances, normality of distribution?

4)      MM section: there is no any information about PCA procedure or correlation coefficients, but authors present results of these analysis.

5)      I suggest to used literature about nitrogen fertilization listed below:

Mastalerczuk G., Borawska-Jarmułowicz B., Kalaji H.M., Dąbrowski P., Gozdowski D. 2017: Some physiological parameters, biomass distribution and carbon allocation in roots of forage grasses growing under different nitrogen dosages. Chiang Mai Journal of Science 44(4): 1295-1303.

Mastalerczuk G., Borawska-Jarmułowicz B., Kalaji H.M., Dąbrowski P., Paderewski J. 2017: Gas-exchange parameters and morphological features of festulolium (Festulolium braunii K. Richert A. Camus) in response to nitrogen dosage. Photosynthetica 55 (1): 20 – 30.

Reviewer 2 Report

Comments and Suggestions for Authors

Dear Authors, 

Its nice manuscript with very attractive research area. I found some little mistake and give you following suggestions to improve it. 

>Different Nitrogen forms in title is not attractive for the reader, please change this word if possible..

>Just after the introduction started with the experiment detail, if author can add the hypothesis of the research or objective it can help to improve the abstract. 

>keyword should not take from the title or abstract part. Need to change and improve them.

> In Material and Method overview of site and experimental material authors didn't mention the year. 

> In Material and Method Experimental design, authors mentioned about the number of trees, treatments and replication but didn't mentioned about the experimental layout design either split plot or factorial or simple RCBD. 

> I have another suggestion for author, in line 131 about table 1 author mentioned the treatment name with A0, A1, A2 and so on, my suggestion is that if author can use the ration in subscript of A to mentioned with each treatment name it can help reader not come back and forth for understanding about the treatment what is A5 or A4 or anyone. Author can disagree with my this comment but its my suggestion to improve it. 

>Line 134 June 15 and November 15 which year same year or different years. If the same year, is it replicate by year too?

>Line 153 authors should mention the complete name of the instrument with company name. 

>In Line 183 ETIF3H is internal control. Why chose this one should mentioned the reference or citation for base of this Housekeeping or Internal control. 

>Line 189 Excel 2007, i think it's too old version why? 

>Figure 1A I am confuse with Diameter unit. Is it mm or inches in diameter? Author should correct by double check with the measurements. 

> Figures Bar color box should mention on the top of the figure its look little odd at the bottom of the figure. 

>Figure 3 Bar color are super dark in black should light them little bit. 

> Figure 6 The expression level should mentioned in fold change units for the expression of each genes. I didn't understand the figure legends and axis titles about the genes. 

> I don't after the discussion conclusion should be in separate heading or same heading?

If you follow these comment i think it will help to improve your manuscript for readers and journal point of view. 

Reviewer 3 Report

Comments and Suggestions for Authors

All my comments on the work entitled "Physiological and molecular responses of Camellia oleifera seedlings under different nitrogen forms" are in the pdf

Reviewer 4 Report

Comments and Suggestions for Authors

This study investigated the effects of different nitrogen forms on the physiological and molecular responses of Camellia oleifera seedlings. While the manuscript is generally well-written, there is a need for more detail in the ‘Materials and Methods’ section and for clarification and correction in the ‘Results’ section, see the specific comments below.

Lines 109-110: You mentioned the use of three varieties in the study, yet the ‘Results’ section did not mention anything about these varieties. Were there any differences among the varieties?

Line 121-123: What was the experimental design? Please provide more details regarding how the experiment was set up and the layout used.

Line 122: Were these 200 seedlings all from the same variety?

Line 146-147: It is not clear how the plants were selected.

Line 154: Why did not you randomly select the plants?

Line 200-202: This statement is not true. From Figure 1A, treatments A1, A2, and A4 were not significantly different in height.

Line 229: Did you mean to say “two” instead of “three” mixed nitrogen treatment groups?

Line 239: From Figure 2B, there was no significant difference between A5 and A0 on root projection area. Please check similar statements throughout the 'Results' section.

Comments on the Quality of English Language

English language is fine.

Round 2

Reviewer 3 Report

Comments and Suggestions for Authors

The authors of the manuscript entitled 'Physiological and molecular responses of Camellia oleifera seedlings to Varied Nitrogen Sources' made the requested adjustments.